# The High-Efficiency Design Method for Capacitive MEMS Accelerometer

**DOI:** 10.3390/mi14101891

**Published:** 2023-09-30

**Authors:** Wen Liu, Tianlong Zhao, Zhiyuan He, Jingze Ye, Shaotong Gong, Xianglong Wang, Yintang Yang

**Affiliations:** 1School of Microelectronics, Xidian University, Xi’an 710071, China; 22111213773@stu.xidian.edu.cn (W.L.); zyhe111111@outlook.com (Z.H.); 23111213537@stu.xidian.edu.cn (J.Y.); 23111213666@stu.xidian.edu.cn (S.G.); xlwang_312@stu.xidian.edu.cn (X.W.); ytyang@xidian.edu.cn (Y.Y.); 2State Key Laboratory of Crystal Materials, Shandong University, Jinan 250100, China

**Keywords:** capacitive MEMS accelerometer, high-efficiency design method, FEM, orthogonal design, PSO algorithm

## Abstract

In this research, a high-efficiency design method of the capacitive MEMS accelerometer is proposed. As the MEMS accelerometer has high precision and a compact structure, much research has been carried out, which mainly focused on the structural design and materials selection. To overcome the inconvenience and inaccuracy of the traditional design method, an orthogonal design and the particle swarm optimization (PSO) algorithm are introduced to improve the design efficiency. The whole process includes a finite element method (FEM) simulation, high-efficiency design, and verification. Through the theoretical analysis, the working mechanism of capacitive MEMS accelerometer is clear. Based on the comparison among the sweep calculation results of these parameters in the FEM software, four representative structural parameters are selected for further study, and they are *l_e_*, *n_f_*, *l_f_* and *w_PM_*, respectively. *l_e_* and *l_f_* are the length of the sensing electrode and fixed electrode on the right. *n_f_* is the number of electrode pairs, and *w_PM_* is the width of the mass block. Then, in order to reduce computation, an orthogonal design is adopted and finally, 81 experimental groups are produced. Sensitivity *S_V_* and mass *M_a_* are defined as evaluation parameters, and structural parameters of experimental groups are imported into the FEM software to obtain the corresponding calculation results. These simulation data are imported into neural networks with the PSO algorithm. For a comprehensively accurate examination, three cases are used to verify our design method, and every case endows the performance parameters with different weights and expected values. The corresponding structural parameters of each case are given out after 24 iterations. Finally, the maximum calculation errors of *S_V_* and *M_a_* are 1.2941% and 0.1335%, respectively, proving the feasibility of the high-efficiency design method.

## 1. Introduction

A MEMS accelerometer transforms the acceleration of an object into an electrical signal, realizing the measurement of acceleration through detecting the electrical one. This device has extensive applications in aerospace, automobiles, seismic surveys and other areas, for it has a compact structure and high precision compared with the traditional type [1,2,3,4]. Compared with a traditional accelerometer, the MEMS type has great advantages on integration and intelligence, which extremely expand its application scenarios.

In general, MEMS accelerometers are divided into piezoresistive, capacitive and piezoelectric types, and so on [5,6,7,8]. As one of the most widely used accelerometers, a piezoresistive MEMS accelerometer takes the piezoresistive effect of a semiconductor as the working principle. Khir et al. [9] reported a low-cost and high-sensitivity CMOS polysilicon thin film piezoresistive accelerometer. A piezoelectric MEMS accelerometer generally utilizes a piezoelectric film, which could produce an electrical charge when deforming. Li et al. [10] designed a novel ZnO film piezoelectric MEMS accelerometer with a composite cantilever beam, significantly improving the sensitivity of the device to 4.5 times that of the regular one. The capacitive MEMS accelerometer is superior with its high accuracy, strong environmental adaptability and excellent dynamic response [11], thus its relative design research, such as material, process and structure, has attracted much attention [12,13,14,15].

As for the structural design of MEMS, most research focuses on a new structure and the lack of a systematic design method [16,17,18,19]. An ordinary design method deeply relies on personal experience, which means low efficiency and inaccuracy. However, those important evaluation criteria, such as measurement accuracy and sensitivity, are mainly decided on the structure of the accelerometer [20,21]. In recent years, the high-efficiency design methods have surged in demand with computational load growing, and the methods proposed at present are developed around the FEM and intelligence algorithm. Zhang et al. [22,23] has conducted many studies in the MEMS optimization area based on the research of Zhou et al. [24,25]. They primarily designed a MEMS simulation tool, which was developed based on global genetic algorithms and local gradient-based refinement. And to improve the performance of the optimal design and synthesis, they also put forward a genotype representation method on the basis of previous research. Wang et al. [26] described a novel, semiautomated design methodology based on a genetic algorithm (GA), which could design and optimize MEMS devices comprising freeform geometries. Finally, an improvement in product sensitivity by 141% and bandwidth by 100% was realized in their work. Li et al. [27] achieved a high sensitivity single-axis comb drive MEMS accelerometer design through the optimization of the gap ratio between anti-finger and finger, finding that a gap ratio of 3.44 led to the ideal sensitivity. Shayaan et al. [28] presented a systematic design method based on the combination of computer experiments (DACE) and Gaussian process (GP) modelling for two degree-of-freedom (2-DoF) capacitive MEMS accelerometer structural optimization, and the maximum prediction accuracy error of five output responses is less than 0.027 in theory.

However, the calculation velocity could be improved further. The particle swarm optimization (PSO) algorithm has advantages in the multi-objective optimization with less computational effort [29], and it has been widely applied in the structural optimization area in recent years. Li et al. [30,31] combined this method with a liquid lens for acoustic pattern control and multi-matching layer of an ultrasonic transducer, and the relative errors of the former are all within 1.0%, verifying the effectiveness of the design method. Chen et al. [32,33] proposed an optimization design strategy based on the PSO algorithm and finite element method (FEM), and the experimental results are in high agreement with the designed ones, realizing the high-performance design ultrasonic transducer. This optimization method has the potential to be applied to the structural optimization of a capacitive MEMS accelerometer, as it is well combined with the transducer design. In this research, the PSO algorithm method is induced to improve the design efficiency of a capacitive MEMS accelerometer. The highlights could be summarized as: (1)The FEM simulation offers calculation data used in the intelligence algorithm, and its sweep results provide accordance with the structural parameters selection.(2)The orthogonal design is induced to condense the calculation, which helps to avoid computational complexity and ensure accuracy at the same time.(3)The high efficiency design method is developed under the PSO algorithm, and the verification of three typical cases indicates the feasibility of the method.

This article picks a typical capacitive MEMS accelerometer with a comb structure as the research object and proposes an intelligence method intergrading PSO algorithm with FEM simulation and orthogonal design, making the design and optimization process more reasonable. In Section 2, the model, working mechanism and FEM method of the MEMS accelerometer are introduced. And Section 3 explains the selection of structural parameters and details of optimization method. Finally, the verification results and relative conclusions are given in Section 4 and Section 5. In the traditional design method, the structural parameters are determined according to the results of all parameter combinations, which are extremely time-consuming and not systematic. The calculating times of three cases are 26.089477, 24.959684 and 25.183701 s, which are far less than for the traditional way. And the final error of the proposed method is below 1.2941%. Above all, the main advantages of the proposed design method are that it is time-saving and has high accuracy.

## 2. Simulation and Analyzation of Capacitive MEMS Accelerometer

### 2.1. Physical Model

The structure of the capacitive MEMS accelerometer is given in Figure 1. The accelerometer generally contains a movable mass block with sensing electrodes and etch holes, fixed electrodes and a folding spring. As we can see, the adjacent electrodes form a plane capacitor, so the whole device could be regarded as several capacitors connected in parallel [34,35]. In practical work, the fixed electrodes are imposed with voltage, and the sensing electrodes move driven by a spring under the effect of acceleration. The variation in the gap between adjacent electrodes leads to the variation in capacitances, which further changes the output voltage. In this way, the mechanical signal is transformed into an electrical one. 

The structural parameters are marked in Figure 1. *l_f_*, *l_e_* and *l_ovrlp_* are the length of the fixed electrode on the right, sensing electrode and the overlapping part of adjacent electrodes. *w_sp_*, *l_sp_*, *w_conn_* and *d_sp_* represent the width, length, connected width and spacing of the folding spring. And *w_PM_*, *w_eh_* and *w_f_* are the width of the mass block, etch hole and electrodes, respectively. *n_f_* means the number of electrode pairs. As shown in Equations (2) and (5), *d*_0_ influences the difference capacitance. *l_f_*, *w_PM_*, *w_eh_* and *w_f_* determine the mass *M* of movable components. *l_ovrlp_*, *l_f_* and *l_e_* possibly influence the output of electrodes, while *w_sp_*, *l_sp_*, *w_conn_* and *d_sp_* may influence the elastic coefficient of the folding spring. Thus, 12 structural parameters are ascertained for the primary research in total.

### 2.2. Working Mechanism

The MEMS accelerometer is a linear electromechanical system, which means its output has a linear relationship with acceleration theoretically. Its basic structure and working principle are shown in Figure 1. The output voltage *V_out_* is directly affected by the variation in capacitance, which is controlled by the displacement of sensing electrodes, respectively. Here are formulas of related parameters:(1)$$x=\frac{Ma}{K}$$
(2)$$C_{0}=\varepsilon \frac{A}{d_{0}}$$
(3)$$C_{1}=\varepsilon \frac{A}{d_{0}+x}\approx \varepsilon \frac{A}{d_{0}}(1-\frac{x}{d_{0}})=C_{0}\left(1-\frac{x}{d_{0}}\right)$$
(4)$$C_{2}=\varepsilon \frac{A}{d_{0}-x}\approx \varepsilon \frac{A}{d_{0}}(1+\frac{x}{d_{0}})=C_{0}\left(1+\frac{x}{d_{0}}\right)$$
(5)$$\Delta C=-2C_{0}\frac{x}{d_{0}}=-2\varepsilon \frac{AMa}{d_{0}^{2}K}$$
where *a* is the acceleration of the device; *x* and *M* represent the displacement and the mass of proof mass, respectively; *K* is the elastic coefficient of the spring; ε is the dielectric constant; and *C*_0_, *C*_1_ and *C*_2_ are the capacitance values formed between electrodes under different working conditions. These formulas listed above are suitable to single electrode pair. In practice, the total results need to multiply the number of electrode pairs *n_f_* on this basis.

As shown in Figure 1b, each sensing electrode is put between the adjacent fixed electrodes. Without the effect of acceleration, equal capacitances *C*_0_ are formed when the fixed electrodes are subjected to opposite voltages ±V. It is easy to see from Equation (2) that the value of the capacitance is directly related to electrodes’ spacing *d*_0_ and the overlapping area *A* of adjacent electrodes. With the influence of acceleration, the capacitances are gradually changing into *C*_1_ and *C*_2_, respectively, resulting in the difference value ∆*C* between the adjacent equivalent capacitors, and their expressions are listed as (3)–(5). Actually, the final output voltage signal is proportional to the difference value ∆*C*, and this value is further proven to be proportional to acceleration. Thus, the output voltage *V_out_*, which is gained from the fixed electrodes, will change with acceleration correspondingly, and the changing amplitude is directly reflected by the displacement of sensing electrodes. 

### 2.3. FEM Model

And to speed up the calculation, half of the accelerometer is selected for further analysis based on its symmetry. From the working mechanism mentioned above, the solid mechanics and static electricity physical fields are selected. The material type is Polycrystalline silicon. Then, according to the working principle of the accelerometer, the corresponding boundary conditions are imposed. 

The anchors of the fixed electrodes are applied with fixed constraints, while the sensing electrodes are movable and applied with voltage ±2.5 mV, respectively. And the body load is applied to realize the acceleration effect of the whole device, and the value is defined as the product of acceleration and whole mass. In practical application, the MEMS accelerometer must be combined with an amplification circuit. To equate the effect of the circuit, the output voltage is amplified by 1000 times in data processing. A steady-state analysis is introduced to calculate the corresponding displacement and voltage results.

In order to obtain the optimum design parameters, it is necessary to investigate the influence of each parameter on the performance of the accelerometer. Some parameters, such as the thickness of the electrodes, may have an influence on the mass *M* or the spring coefficient *K* in the primary theory analysis, but they are found to be non-influential on the output voltage *V_out_* using the FEM calculation. Therefore, finite element software is introduced to assist in parameter selection. COMSOL 6.1 provides a parameter sweep analysis, which allows the influence of different parameters to be calculated directly. The initial setup of the relevant parameters is listed in Table 1.

## 3. High-Efficiency Design Method of MEMS

### 3.1. Structural Parameters Analysis

The output voltage *V_out_* directly reflects the performance of the capacitive MEMS accelerometer. In addition, for a lightweight design, the overall mass *M_a_* is selected as the other performance evaluation parameter. Displacement *x* is introduced to test the reasonableness of the structure, for there is limited movable space to mass block. It is quite difficult to realize the sweep calculation covering all relevant parameters at the same time. Therefore, pick out the most suitable structural parameters to reduce the computation first.

The sweep results are shown in Figure 2, and other relevant results are attached in the Appendix A. As for the output voltage *V_out_*, *w_f_*, *w_eh_* and *d_sp_* have unobvious changing tendencies. As for *M_a_*, *d*_0_ and *l_ovrlp_* remain constant; *l_sp_*, *w_sp_*, *d_sp_* and *w_conn_* also have relatively flat changing trend. Thus, 4 representative structural parameters *l_e_*, *n_f_*, *l_f_* and *w_PM_* are selected for further study. It is shown that the four parameters variously influence performance. The displacement results are controlled beneath 0.1 μm. And the influences on *V_out_* behave in different trends. *l_f_* and *l_e_* show nonlinear trends compared with *w_PM_* and *n_f_*, and *n_f_* has the most obvious impact. From Figure 2c, these parameters have linear correlations with *M_a_*.

To ascertain the scales and steps of all relative parameters is quite necessary before sweep calculation and analysis. To make sure structural rationality of capacitive MEMS accelerometer, the determination of parameters’ scale depends on the initial structural size of the model in this article. And to show the regulation of various parameters’ influence clearly, their sweep scales are determined to be as large as possible. A good example is that the initial gap between adjacent electrodes is 1 μm, so to avoid the direct touching between the electrodes and make the changing regulation as detailed as possible, the initial scale of the electrode spacing *d*_0_ is established as [0.1, 1.4] μm and the changing step is 0.1 μm. In the same way, calculation scales of other parameters are decided. The orthogonal design can condense calculations through generating representative experimental groups, thus the design method is combined with an orthogonal design in this article.

In the traditional design method, the values of structural parameters are determined based on the calculation results obtained from various parameter combinations. However, there are still 9^4^ kinds of combinations after parameter selection, which is almost impossible to realize using FEM analysis because the calculation amount is too large. Therefore, the highly efficient design method combining the PSO algorithm with FEM simulation and orthogonal design is proposed. And its design process is shown in Figure 3.

### 3.2. Orthogonal Design

Various parameters are given with different scales and steps, and each is divided into nine groups; a detailed parameter list is provided in Table 2. The scales of *n_f_* and *l_f_* are [13, 21] and [111, 119] μm, respectively, and their steps are both 1 μm. As for *l_e_* and *w_PM_*, the step is 2 μm and their scales are [106, 122] and [100, 116] μm. In the following research, the output voltage *V_out_* is replaced by sensitivity *S_V_*, which is defined as the variation in *V_out_* divided by the variation in acceleration. *S_V_* accurately reflects the device’s response severity to external stimuli.

The orthogonal design is a practical approach to select the most representative groups. This step is accomplished using the SPSS orthogonal design toolbox. To minimize the number of calculation groups, the interactions between different parameters are ignored. The experimental levels are set as *M_a_* and *S_V_*, while the structural parameters are considered variables. These experimental groups are then imported into COMSOL, and the performance of each group is calculated. There are totally 81 sets of data, and Table 3 shows some of the experimental results. The maximum and minimum of *S_V_* are 0.4641 and 0.3097 mV/g, and the corresponding values of *M_a_* are 3.6602 × 10^−10^ and 2.7690 × 10^−10^ g.

### 3.3. PSO Algorithm

All experimental data are then imported into the Artificial Neural Network (ANN) toolbox, and the PSO algorithm is utilized in structural optimization design. Furthermore, due to the small size and light weight of the MEMS, the value of *M_a_* is quite small, thus the mass data used for the calculation are magnified 10^10^ times. The ANN is a computation model that functions similarly to nerve cells in the human brain. The PSO algorithm utilized in this study begins with random solutions and searches for the optimal solution through iteration, adopting the inertia weight adaption mechanism [36,37,38,39]. The training model will keep calculating until the calculated values are quite close to expectation within the relatively limited calculations. 

The normalized optimality criteria of the PSO algorithm can be expressed as
(6)$$J=\alpha \left(\frac{S_{V}-{S_{V}}_{exp}}{{S_{V}}_{max}-{S_{V}}_{min}}\right)+\beta \left(\frac{M_{a}-{M_{a}}_{exp}}{{M_{a}}_{max}-{M_{a}}_{min}}\right)$$
where ${S_{V}}_{exp}$ and ${M_{a}}_{exp}$ represent the expectation of sensitivity and overall mass; ${S_{V}}_{max}$, ${S_{V}}_{min}$, ${M_{a}}_{max}$ and ${M_{a}}_{min}$ represent the maximum and minimum values of corresponding parameters; and α and β are the weight coefficients.

For some practical applications, the sensitivity *S_V_* is expected to be as high as possible, while the mass *M_a_* is expected to be relatively small. However, it is difficult to achieve high levels of both performances at the same time. It is necessary to find the balance between the two parameters in practical design. The high-efficiency design method provides great convenience for the definite optimization goals.

In order to test the effectiveness of the design method in the face of different working requirements, three typical cases are proposed here, and their details are listed in Table 4, where the expected value means the goal to be achieved and the calculated value means the result given by the PSO algorithm. The two performance parameters are given different weights to represent various working conditions, and each case is given a corresponding optimization objective. Their weights and expected values are randomly selected in the data range, and the selection in this article is just an example for testing feasibility. For case A, the optimized goals are determined as *S_V_* = 0.4 mV/g and *M_a_* = 3.2 g, and they are granted equal weights. Case B focuses on the optimization of *S_V_*, so its weight is defined to be larger than *M_a_* and expected values are raised based on case A at the same time. And case C is set to the contrary. The results of 24 learning iterations are shown in Figure 4. It is shown that the calculated *S_V_* for three cases are 0.3971, 0.4481 and 0.3246 mV/g approximately, while their calculated *M_a_* are around 3.2056, 3.5410 and 2.8467. The stability of the results indicates the feasibility of the structural optimization method applied to the MEMS accelerometer. Most importantly, with the help of the PSO algorithm, the optimization consuming times of three cases are 26.089477, 24.959684 and 25.183701 s. For the traditional optimization method, it requires the calculation of all relevant structure parameter combinations, and the time taken could be several hours or longer. Here, a highly efficient optimization design method is fully proposed.

## 4. Verification and Discussion

To validate the effectiveness and accuracy of the method, the structural parameters obtained from the high-efficiency method are imported into COMSOL. The induced voltage outcomes *V_out_* are calculated to gain the sensitivity *S_V_*. The high-efficiency design method gives out the theoretical structural parameters of each case, and they are *n_f_* = 18, *l_f_* = 114.6832 μm, *l_e_* = 107.2724 μm and *w_PM_* = 101.2724 μm, respectively, for case A. Then, these data are used in the FEM model, and the corresponding calculation results are *S_V_* = 0.3967 mV/g and *M_a_* = 3.2056 g. Cases B and C are implemented in this way as well. 

The computed results are depicted in Figure 5 and Figure 6 and Table 5. From Figure 6a, it is easy to know that case B has the maximum displacement as it has the maximum whole mass *M_a_*, and the displacement value at 50g acceleration is less than 0.04 μm. The displacement results signify the feasibility of the structure as they are all controlled within limited ranges. However, the output voltage *V_out_* of cases A and C are not symmetrical about a static point in Figure 6b, which might be a result of them having less electrode pairs compared with case B. In this paper, this nonlinear effect is neglected, and *S_V_* is selected for study only. 

The relative error in Table 5 is defined as
(7)$$Relative\;error=\left|\frac{Calculated\;value-Simulation\;value}{Calculated\;value}\right|\times 100\%$$

The comparison results between calculation and simulation are listed in Table 5. The calculated value is the final outcome given by the PSO algorithm, and the simulation value means the outcome gained from the FEM model. And the table shows that the maximum relative errors of *S_V_* and *M_a_* are 1.2941% and 0.1335%, respectively. In addition, there are differences among the errors of various cases, which may relate to their different optimization goals and the assigned different weights. In general, the simulation values are quite close to the expected one, indicating the feasibility of the high-efficiency method. 

## 5. Conclusions

To overcome the massive calculation and inaccuracy in traditional design, a high-efficiency design method of the MEMS accelerometer is proposed. This design method is developed based on the combination of an FEM sweep analysis, orthogonal design and the PSO algorithm. And the final results testify to the feasibility of the proposed method. The main conclusions are as follows:(1)The physical model and working mechanism of the capacitive MEMS accelerometer is discussed in detail. The theory analysis and sweep analysis provide dependence for the primary structural parameters selection, and the FEM calculation results provide data for the PSO algorithm.(2)The high-efficiency design method is proposed based on the combination of the orthogonal design and PSO algorithm. The orthogonal design is introduced to reduce the calculation amount. And the PSO algorithm greatly saved the time taken for designing compared with the traditional way. Structural parameters *l_e_*, *n_f_*, *l_f_* and *w_PM_* were chosen for further research depending on the FEM sweep analysis.(3)To verify the accuracy of the design method, the calculated structural parameters of three cases were imported into COMSOL. And the corresponding FEM results indicated the feasibility and effectiveness of the method as the calculation errors of *S_V_* and *M_a_* were no more than 1.2941% and 0.1335%.

## Figures and Tables

**Figure 1 micromachines-14-01891-f001:**
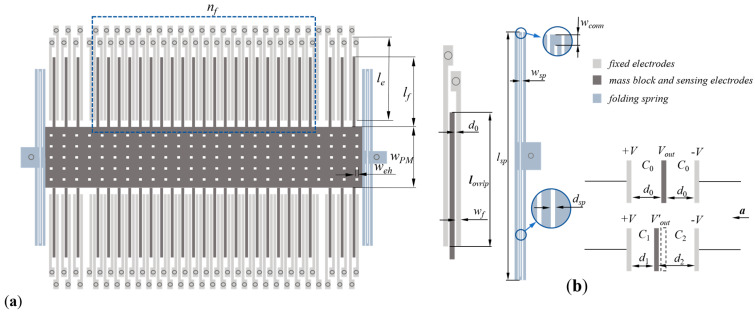
Structure of MEMS accelerometer and its working principle: (**a**) structure; (**b**) working principle.

**Figure 2 micromachines-14-01891-f002:**
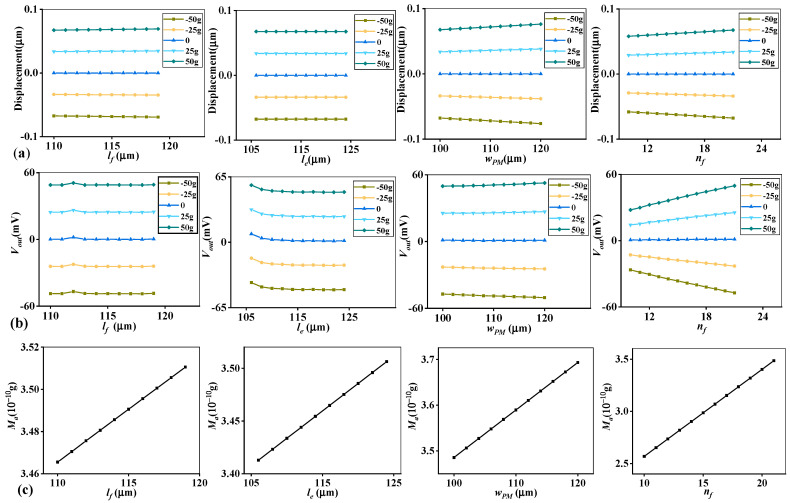
Sweep results of four selected parameters: (**a**) displacement; (**b**) induced voltage; and (**c**) overall mass *M_a_*.

**Figure 3 micromachines-14-01891-f003:**
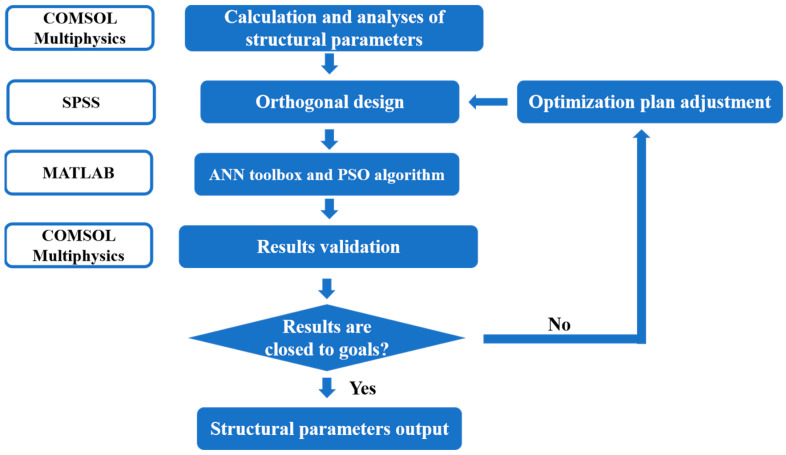
Design process of high-efficiency method.

**Figure 4 micromachines-14-01891-f004:**
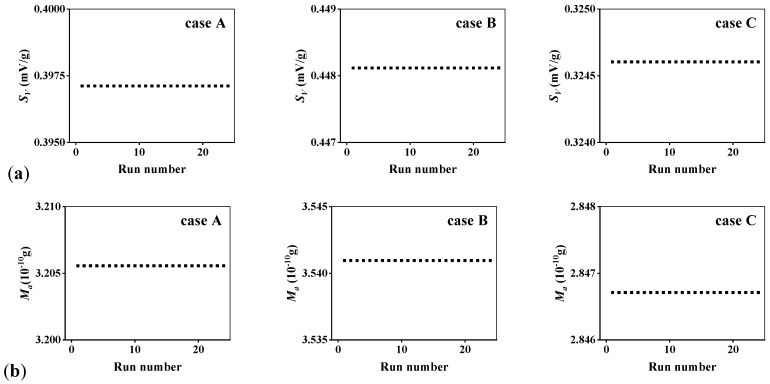
Results of three cases optimized using high-efficiency method: (**a**) sensitivity *S_V_*; (**b**) overall mass *M_a_*.

**Figure 5 micromachines-14-01891-f005:**
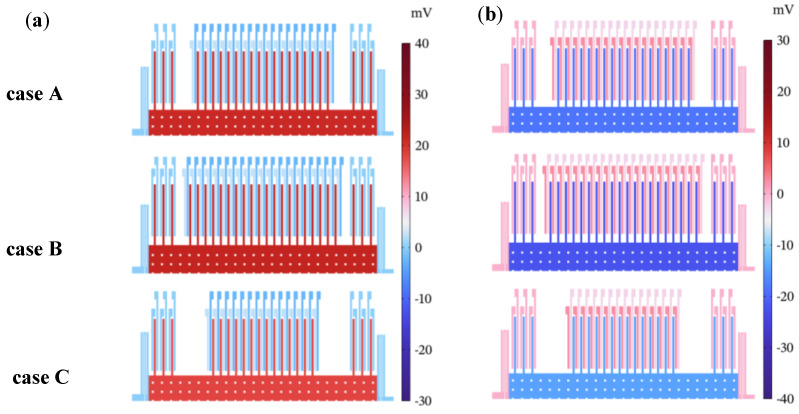
Voltage distributions of three cases in COMSOL: (**a**) a = 50 g; (**b**) a = −50 g.

**Figure 6 micromachines-14-01891-f006:**
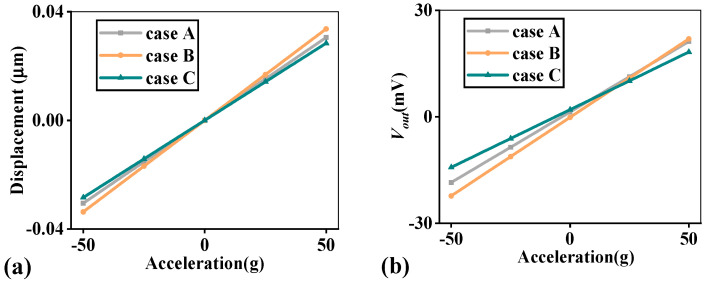
Simulation results of three cases: (**a**) displacement; (**b**) output voltage *V_out_*.

**Table 1 micromachines-14-01891-t001:** Initial setup of parameters in FEM simulation.

Parameter	Initial Setup
Acceleration	−50 g, −25 g, 0, 25 g, 50 g
*w_eh_*	4 μm
*d* _0_	1 μm
*n_f_*	21
*l_sp_*	280 μm
*w_sp_*	2 μm
*l_ovrlp_*	104 μm
*l_f_*	114 μm
*l_e_*	120 μm
*w_PM_*	100 μm
*w_f_*	4 μm
*w_conn_*	4 μm
*d_sp_*	1 μm

**Table 2 micromachines-14-01891-t002:** Scale and step of 4 selected parameters.

Parameter	*n_f_*	*l_f_*	*l_e_*	*w_PM_*
*Scale* (μm)	[13, 21]	[111, 119]	[106, 122]	[100, 116]
*Step* (μm)	1	1	2	2

**Table 3 micromachines-14-01891-t003:** Part data of the experimental groups.

*n_f_*	*l_f_* (μm)	*l_e_* (μm)	*w_PM_* (μm)	*S_V_* (mV/g)	*M_a_* (10^−10^ g)
16	111	112	106	0.3671	3.0977
18	116	106	116	0.4137	3.3592
18	112	114	114	0.4118	3.3574
17	117	120	116	0.3948	3.3448
16	119	120	110	0.3703	3.2067
16	116	108	112	0.3739	3.1639
20	113	122	110	0.4416	3.5244
19	117.00	106.00	102.00	0.4140	3.2991
14	112.00	106.00	106.00	0.3308	2.9155

**Table 4 micromachines-14-01891-t004:** Optimization cases under different weights.

Case	*n_f_*	*l_f_* (μm)	*l_e_* (μm)	*w_PM_* (μm)	Parameter	Weight	Expected Value	Calculated Value
A	18	114.6831	107.2723	101.2724	*S_V_* (mV/g)	0.5	0.4	0.3971
					*M_a_* (10^−10^ g)	0.5	3.2	3.2056
B	20	119	117.1264	111.1264	*S_V_* (mV/g)	0.7	0.45	0.4481
					*M_a_* (10^−10^ g)	0.3	3.5	3.5410
C	14	111	106.0934	100.0934	*S_V_* (mV/g)	0.3	0.33	0.3246
					*M_a_* (10^−10^ g)	0.7	2.8	2.8467

**Table 5 micromachines-14-01891-t005:** Comparison between calculated value and simulation value.

Parameter	Case	Calculated Value	Simulation Value	Relative Error (%)
*S_V_* (mV/g)	A	0.3971	0.3970	0.0252
	B	0.4481	0.4423	1.2941
	C	0.3246	0.3247	0.0308
*M_a_* (10^−10^ g)	A	3.2056	3.2057	0.0031
	B	3.5409	3.5410	0.0028
	C	2.8467	2.8505	0.1335

## Data Availability

The data used to support the findings of this study are available from the corresponding author upon request.

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
