# Peer review of "The High-Efficiency Design Method for Capacitive MEMS Accelerometer"

_micromachines, 2023, doi:10.3390/mi14101891_

Round 1
Reviewer 1 Report
This manuscript reports a design method of a capacitive MEMS accelerometer using particle swarm optimization (PSO) algorithm and finite element method (FEM) simulations. This manuscript can be enhanced based on the following issues:
1.-The abstract must include a description of all the parameters. For instance, the following parameters are not described: le, nf, lf, wpm.
2.-The introduction section should add the main research problem. In addition, this section must consider the main advantages and limitations of design methods of MEMS accelerometers reported in the literature. Also, the discussion of these methods should be improved.
3.-What are the main advantages and drawbacks of the proposed design method?
4.- The title "Simulation and analyzation of MEMS" of the second section should be enhanced.
5.-The authors should enhance the description and schematic views of the physical model of the MEMS accelerometer.
6.-The description of the FEM model must incorporate more information on the boundary conditions, constraints, and analysis types. The authors should add the analysis types used in the proposed FEM models. Can the proposed FEM model include modal and electrothermal analyses?
7.-The authors must improve the discussions on the behavior of the results of Figures 2, 4, and 5 and Tables 1-4.
8.-What are the future research works?
9.-The authors should consider discussions on the influence of the damping and temperature on the accelerometer performance.
10.-The conclusion section is weak. This section must be enhanced.
11.-The authors could include more recent references between 2021 and 2023.
The English grammar can be enhanced.
Reviewer 2 Report
The manuscript presents a design of capacitive MEMS accelerometers based on FEM simulation with some optimization. The novelty and significance of this work are not clear. The manuscript doesn't clearly state the current issues of the MEMS accelerometer design and what challenges could be resolved in this work. There is no comparison of the presented design method, achieved device performances with other works.
Minor editing of English is required.
Reviewer 3 Report
The authors propose a high-efficiency design method for capacitive MEMS accelerometers based on the combination of FEM sweep analysis, orthogonal design, and PSO algorithm. The proposed design method of the MEMS accelerometer is verified to be accurate with a maximum design error of SV and Ma of 1.2941% and 0.1335%, respectively. Overall, the work is relatively systematic and of value. However, the novelty is moderate, and some issues need to be resolved:
1. As far as I am concerned, the authors should research the parameters that have significant impacts on the output, while in section 3.1, d0 and wsp are excluded. Please explain it.
2. The authors should explain how they chose the weight in Table 3.
3. As a high-efficiency design method, the authors are recommended to present the setup, optimization consuming time, etc.
4. There are a lot of typos (like two Tables of 1, and the units of Ma in Table 4); a thorough proofreading of the format is recommended.
Round 2
Reviewer 1 Report
The authors have addressed all the reviewer's comments.
English grammar is acceptable.
Reviewer 2 Report
The authors have addressed all the comments
Minor editing